# A study on the potential of higher education in reducing carbon intensity

Qin Yuan[1☉], Ruiqi Wang[2☉], Huanchen Tang[3]*, Xin Ma[4], Xinyue Zeng[5]

**1** School of Art, Southeast University, Nanjing, China, **2** Nanjing Water Group CO., LTD, Nanjing, China, **3** College of Fashion and Art Design, Donghua University, Shanghai, China, **4** College of Civil Engineering, Nanjing Forestry University, Nanjing, China, **5** Massey College, Nanjing University of Finance and Economics, Nanjing, China

☉ These authors contributed equally to this work.
* tanghc63@gmail.com

**Data Availability Statement:** All relevant data are within the paper and its Supporting Information files.

**Funding:** The author(s) received no specific funding for this work.

## Abstract

The Chinese government has established definitive goals to reach a "carbon peak" by 2030 and achieve "carbon neutrality" by 2060. Investigating the attainment of these emission reduction objectives while simultaneously fostering regional economic growth and enhancing living standards holds critical importance. This study examines the link between higher education and carbon intensity across China's thirty provincial-level administrative regions, employing fixed effects models on provincial panel data spanning 2001–2020. The findings, validated through robustness tests and a mediation effect model, elucidate the mechanisms by which higher education influences carbon intensity. Notably, the results reveal that enhancing higher education markedly lowers carbon intensity; specifically, a 1% increase in the logarithmic transformation of per capita investment in higher education in a province decreases its carbon intensity by 0.219%. Additionally, higher education's output similarly contributes to reductions in carbon intensity. The influence of higher education on reducing carbon intensity is particularly pronounced in the central and western regions of China. Moreover, higher education facilitates the reduction of carbon intensity through mechanisms such as promoting environmental consciousness, advancing industrial structure, and encouraging technological innovation.

## 1. Introduction

The emergence of global climate warming poses a substantial threat to the attainment of sustainable human development. Climate warming leads not only to the rise in sea levels, and the extinction of species [1], but also exerts long-term impacts on human capital [2] and social productivity [3]. Addressing global climate warming has become a consensus among people worldwide, and effectively mitigating it has become a paramount task for governments globally. Extensive research suggests that the primary factor responsible for global climate warming is the emission of carbon dioxide resulting from human activities [4]. Human activities involve substantial consumption of fossil energy, resulting in significant carbon dioxide emissions. The accumulation of carbon dioxide in the atmosphere leads to the greenhouse effect [5]. As

**Competing interests:** The authors have declared that no competing interests exist.

the world's largest energy consumer, China releases a substantial amount of carbon dioxide through the excessive use of coal, oil, and other fossil fuels, contributing over 30% of the global carbon emissions and ranking first in the world [6]. Meanwhile, China is currently engaged in a comprehensive effort to advance the reduction of carbon emissions, actively assuming the onus of emission cuts, and regarding the abatement of carbon emissions as a necessary national strategic pursuit. The governmental authorities of China have explicitly stated their aspiration to attain the "peak carbon emissions" by the year 2030, and to achieve "carbon neutrality" by the year 2060, thereby pledging to effectuate a reduction in carbon intensity exceeding 65% relative to the levels recorded in 2005 [7].As the largest developing nation globally, China confronts a formidable challenge in balancing emission reduction goals with regional economic development and enhancing the living standards of its population.

Beyond legislative measures, the government effectively mitigates carbon emissions by implementing educational initiatives [8]. The 1992 United Nations Framework Convention on Climate Change (UNFCCC) emphasized the essential role of education in tackling global climate change [9]. Solutions to climate change often hinge on individual citizens, and environmental awareness is crucial for any eco-friendly measure or plan. Education is the prime means to instill this awareness [10]. Higher education, at the core of the educational system, may significantly impact carbon emissions through various channels. From an individual perspective, higher education can lead to a deeper understanding of the harmonious coexistence between humans and nature, promoting social responsibility and environmental consciousness [11]. From a corporate standpoint, higher education can facilitate university-enterprise integration, driving technological innovation, reducing resource consumption, and enhancing resource efficiency to mitigate carbon emissions [12]. At an industry level, higher education fosters the accumulation of human capital, enriches its levels, drives industrial restructuring, and thus influences carbon intensity [13]. Nevertheless, the available research on the correlation between higher education and carbon intensity is currently insufficient or limited in scope. Can higher education indeed lower carbon intensity? What mechanisms underlie the connection between higher education and carbon intensity? Do the relationships between higher education and carbon intensity vary across different regions? Answers to these questions are eagerly awaited.

This study analyzes the relationship between higher education and carbon intensity across thirty provincial administrative regions in China using a fixed-effects model. To strengthen the robustness of our findings, we substituted the core explanatory variables, further validating our conclusions. The analysis then categorizes China into eastern and central-western regions to explore regional disparities in the impact of higher education on carbon intensity. Additionally, we developed a mediation effect model that elucidates the roles of technological innovation incentives, environmental awareness promotion, and industrial structure enhancement in mediating this relationship. Based on these insights, we offer targeted policy recommendations. In comparison to existing studies, this paper makes three primary contributions: First, it proposes a theoretical framework from the perspective of higher education, thereby broadening the environmental governance discourse. Second, it addresses both the spatial variability in higher education and its differential impacts on carbon intensity, considering both higher educational inputs and outputs. Third, it identifies critical mechanisms such as technological innovation, environmental awareness, and industrial upgrading that link higher education and carbon intensity.

The structure of this paper is organized as follows: Section 2 reviews the relevant literature and provides a theoretical analysis. Section 3 details the data and variables used in the study. Section 4 presents the empirical results. Section 5 examines the mechanisms that influence these results. Finally, Section 6 concludes with the study's findings and offers recommendations.

## 2. Literature review and theoretical analysis

### Study on carbon intensity

The assessment of carbon emission efficiency relies on the important indicator known as carbon intensity, which is defined as the ratio of carbon dioxide emissions to GDP. This crucial metric allows for an evaluation of the efficiency of carbon emissions in relation to economic output [14]. The reduction of carbon intensity not only guides the strategic transformation and upgrading of economies but also acts as a crucial prerequisite for curbing energy consumption [15], regulating emission levels [16], and sustaining economic growth [17]. Hence, the identification of factors influencing carbon intensity has become a focal point of research. The study into these factors can be broadly categorized into three dimensions. Firstly, technological progress plays a fundamental role in reducing carbon intensity. Technological advancements enhance the efficiency of social production, facilitate industrial structural upgrading, and thus lead to a decrease in carbon intensity [18]. Particularly in the context of China, domestic research and development, as well as the spillover effects of imported technologies, exert a significant influence on the reduction of carbon intensity [19]. Technological progress not only constitutes the primary contributor to the reduction of carbon dioxide emissions [20] but also serves as a pivotal means for China to achieve its dual carbon goals [21]. Moreover, technological spillover effects further facilitate the reduction of carbon intensity [19]. Secondly, trade openness exerts an impact on carbon emissions and economic development [22]. It represents a crucial factor influencing carbon intensity. On the one hand, international trade openness enables nations engaged in cross-border trade to share or acquire clean energy technologies [23]. On the other hand, trade openness can lead to changes in economic structure and activities, thereby contributing to environmental degradation [24]. Thereby, trade openness exhibits distinct heterogeneous impacts on carbon intensity. For instance, the research conducted by Wang et al. revealed that in high-income and middle-to-low-income groups, trade openness resulted in a reduction of carbon intensity, while in low-income groups, it led to an increase in carbon intensity [25]. The third factor is industrial structural adjustment. Industrial structure reflects the proportion and changes in various industries within a nation or region. The rationalization, transformation, and upgrading of industrial structures not only constitute key factors in reducing carbon emissions [26] but also serve as determinants of the rate and quality of economic growth [27]. Zhang et al. discovered that the increased proportion of China's tertiary industry and economic growth play a vital role in curbing carbon intensity, with economic growth being the primary influencing factor in changes to carbon intensity [28].

### Study on higher education

From an individual perspective, whether it be China's ancient imperial examination system or the contemporary college entrance examination, education serves as a crucial avenue for ordinary citizens to transcend social classes [29]. Presently, although China has implemented compulsory education laws to ensure primary and junior secondary education for all, higher education remains a scarce and distinctive resource. On the one hand, higher education not only emphasizes specialized knowledge and skills but also places significant emphasis on holistic development and humanistic qualities in students. By fostering critical thinking, problem-solving abilities, and innovative thinking, higher education promotes lifelong learning and personal development [30]. On the other hand, higher education provides broader employment opportunities and prospects for career development for students [31]. Through specialized training and learning, students acquire profound professional knowledge and skills, thereby enhancing their income and status in the workplace.

From a societal perspective, higher education exerts a multifaceted impact on the economy. Firstly, higher education endows society with a labor force of exceptional quality. This skilled human capital enhances productivity and competitiveness, thereby propelling economic growth [32]. Secondly, higher education serves as a crucial avenue for nurturing innovative talents and advancing scientific and technological progress, thereby providing support for technological innovation and industrial development [33]. Finally, higher education facilitates greater labor mobility and adaptability, thereby facilitating labor force movement and enhancing labor market efficiency [34].

However, the existing body of literature on the relationship between higher education and carbon intensity is relatively scarce and fails to acknowledge the significant role of education in addressing environmental concerns [8]. Some studies, through the distribution of surveys, have found that individuals with higher levels of education tend to exhibit a stronger sense of social responsibility. They are often more environmentally conscious and inclined to resist high-energy-consuming production and consumption methods [35]. Additionally, educated individuals are more likely to sensibly utilize idle items [36] and employ clean energy products [37]. For instance, they are more inclined to purchase new energy vehicles [38]. An individual's environmental consciousness is often highly correlated with their educational experience [39]. Education that emphasizes environmental friendliness can lead to environmentally friendly behaviors, thereby promoting carbon emission reduction and achieving nature conservation goals [40]. However, the aforementioned literature only provides a microscopic understanding of a potential link between higher education and the environment, without empirically demonstrating the causal relationship between higher education and carbon intensity.

In summary, extensive literature on carbon intensity identifies technological progress, trade openness, and industrial structure as key determinants. Despite this, the link between higher education and carbon intensity remains unexplored. Current research predominantly addresses the significance of higher education to individuals and society, neglecting its potential environmental impacts. While some scholars suggest a possible connection between higher education and environmental outcomes, empirical evidence supporting a causal relationship with carbon intensity is lacking. Consequently, a rigorous scientific assessment of this relationship is crucial.

## Theoretical analysis and research hypothesis

From a behavioral economics standpoint, highly educated individuals typically exhibit a heightened sense of social responsibility. They are more attentive to environmental sustainability and actively oppose lifestyles and consumption habits characterized by high energy use [41]. Such individuals are predisposed to adopting pro-environmental behaviors and demonstrating a robust environmental consciousness [42]. In both professional and personal contexts, they prefer to efficiently utilize idle resources, opt for low-carbon transportation solutions, and use environmentally friendly products. For example, as consumers, educated individuals tend to favor purchasing electric vehicles to mitigate carbon emissions [43]. Furthermore, educated entrepreneurs and managers are more likely to uphold corporate social responsibilities, advocate for sustainable production methods, and minimize the ecological impact of their business operations *[10,44]*.

According to human capital theory, education and training represent crucial investments in human capital [45]. Higher education not only augments workers' professional knowledge and practical skills but also fosters their capacity for innovation and research [46]. This comprehensive enhancement significantly improves their level of knowledge, professional abilities,

and overall competencies, thereby increasing their potential future earnings and job prospects [47]. The educational advancement of the workforce contributes to the accumulation and diversification of human capital, which, in turn, drives the growth of high-tech industries and the evolution of industrial structures. Consequently, this shift leads to the displacement of traditional high-pollution industries by more sustainable high-tech and green sectors, effectively reducing carbon intensity *[44,48]*.

Theoretical economics suggests that higher education not only raises an individual's knowledge level but also generates knowledge spillover effects [49]. This educational process not only benefits the learners themselves but also exerts a positive influence on the wider society [50]. By engaging in both educational and research activities, universities can attract technology-driven enterprises to establish collaborative platforms such as joint laboratories, technology transfer centers, and innovation incubators [51]. These platforms not only enable joint research projects and technological advancements but also facilitate the specialized training of innovative talent [52]. They promote local technological innovation, catalyze the development of emerging clean technologies, improve resource efficiency in production, and ultimately contribute to a reduction in carbon intensity *[44,53]*.

Based on the analysis above, this paper illustrates the discussed mechanisms in Fig 1 and proposes the following hypotheses:

Hypothesis 1: Higher education can significantly suppress carbon intensity.

Hypothesis 2: Higher education reduces carbon intensity through the mechanism of promoting environmental awareness.

Hypothesis 3: Higher education reduces carbon intensity through the mechanism of upgrading industrial structures.

Hypothesis 4: Higher education reduces carbon intensity through the mechanism of incentivizing technological innovation.

## 3. Variable selection and data source

### Variables and data sources

**Dependent variable.** The dependent variable (DV) in this study is carbon intensity, which is measured as the ratio of carbon dioxide emissions to GDP. The calculation is represented by Eq (1):

$$CI = \frac{CE}{GDP} \tag{1}$$

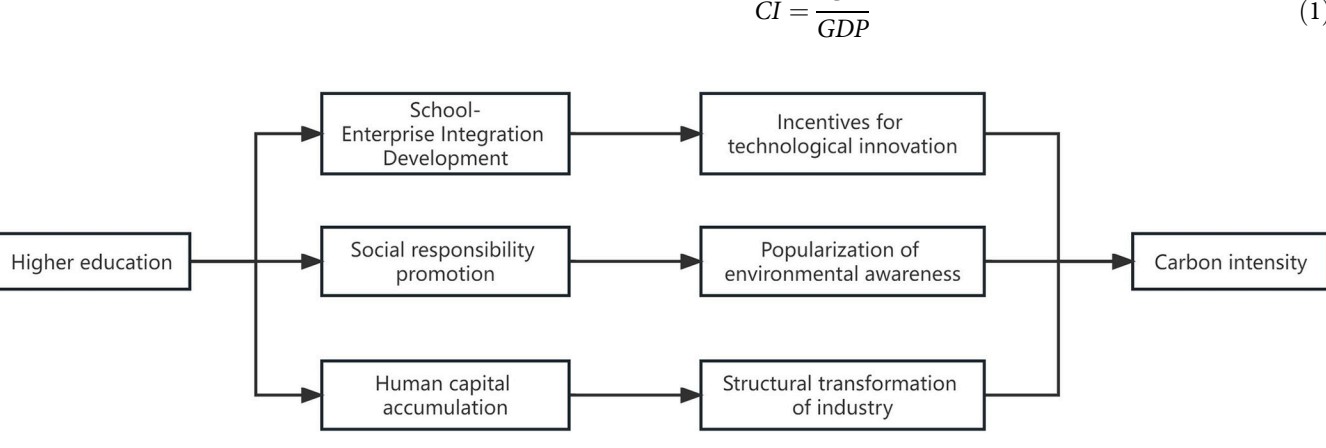

**Fig 1. Mechanistic analysis of higher education and carbon intensity.**

Herein, CI represents regional carbon intensity, CE stands for regional carbon emissions, and GDP denotes regional gross domestic product. The GDP data is sourced from the China Statistical Yearbook. Due to the absence of official carbon dioxide statistics from the Chinese government, regional carbon emissions are meticulously categorized into three parts based on the geographic location of emission sources. The combined carbon dioxide emissions from these parts constitute the total carbon dioxide emissions of the region [54]. Carbon emissions are calculated using Eq (2):

$$CE = CE_1 + CE_2 + CE_3 \qquad (2)$$

$CE_1$ encompasses all direct emissions occurring within the jurisdiction, predominantly arising from intra-jurisdictional energy activities. These emissions are classified as greenhouse gas (GHG) emissions from transportation and buildings, industrial processes, agriculture, land-use change and forestry, as well as waste disposal activities. $CE_2$ pertains to indirect energy-related emissions outside the jurisdiction, primarily stemming from the acquisition of secondary energy sources like electricity, heating, and/or cooling for regional consumption. $CE_3$ encompasses additional indirect emissions resulting from activities within the jurisdiction but not addressed in Part II. This includes all items procured from outside the jurisdiction involved in the production, transportation, and consumption of energy. $CE_3$ also includes indirect emissions resulting from activities within the jurisdiction but occurring outside its boundaries, not covered in Part II. These encompass GHG emissions from the production, transportation, use, and waste disposal of all goods acquired from outside the jurisdiction. Table 1 illustrates the categorization of carbon emissions.

The carbon dioxide data utilized in this study predominantly originates from diverse industry statistical yearbooks. Data pertaining to transportation, construction, and industrial production processes are extracted from the "China Industrial Statistical Yearbook" and assorted statistical yearbooks at different levels. Information regarding agricultural, forestry, and other land use activities is gathered from publications such as the "China Agricultural Statistical Yearbook," "China Animal Husbandry Yearbook," "China Forestry and Grassland Statistical Yearbook," as well as various statistical yearbooks across different levels. Data concerning waste disposal is acquired from the "China Environmental Statistical Yearbook" and multiple statistical yearbooks at different levels. Information regarding purchased electricity, heating, and cooling is derived from publications such as the "China Urban Statistical Yearbook," "China Energy Statistical Yearbook," and various statistical yearbooks at different levels. Energy consumption data pertaining to the energy sector is extracted from the "China Energy Statistical Yearbook" and various statistical yearbooks at different levels. Since the Chinese government exclusively disclosed industry statistical data preceding 2020, data spanning from 2001 to 2020 was employed in this study. Fig 2 depicts the categorization of carbon emissions in China for the year 2010. Direct emissions constitute the primary component of regional carbon emissions, notably with economically advanced regions such as Beijing, Guangdong, and Shanghai emitting a greater quantity of carbon compared to other areas.

**Table 1. Categorization of carbon emissions.**

| Scope 1 | Direct carbon emissions within the jurisdiction | Transportation and construction, Industrial process; Agroforestry and land-use change; Waste disposal |
|---|---|---|
| Scope 2 | Direct carbon emissions outside the jurisdiction | Purchased electricity; Heating and cooling |
| Scope 3 | Other carbon emissions | Production, transportation, use and waste disposal of all items purchased |

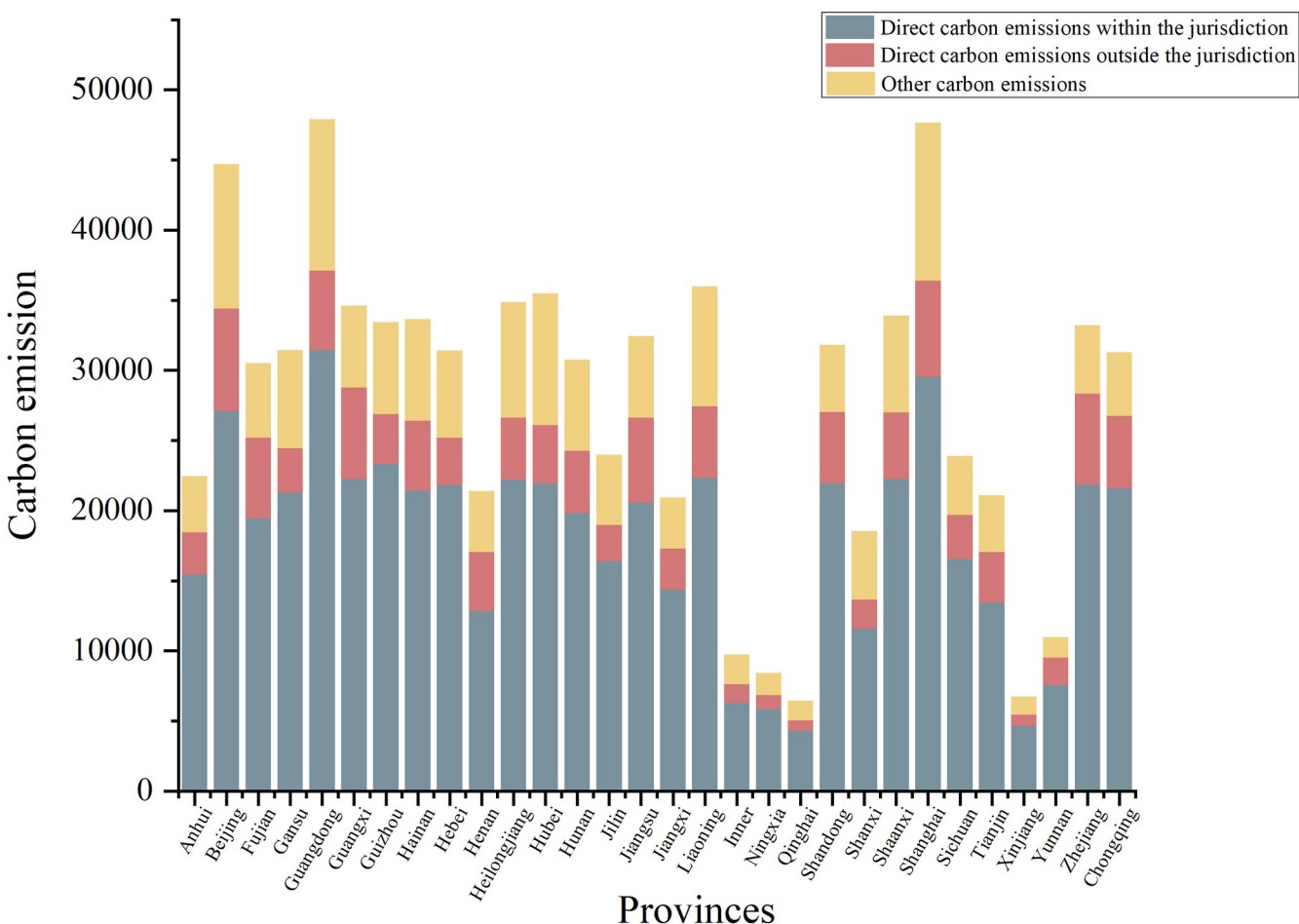

**Fig 2. Categorization of carbon emissions in various Chinese regions.**

**Independent variable.** The independent variable (IV) in this study is higher education, which is evaluated through local investment in higher education. This constitutes a significant indicator that reflects the degree of attention and scale of higher education across provinces and can explicate varying levels of emphasis placed on higher education across different regions. Several measurement methods are commonly employed, including measuring higher education funding as a proportion of GDP [55], utilizing per capita general public budgetary education expenses for higher education [56], and directly employing higher education expenditure by provinces [57]. Given that per capita public budget expenditures on education can mitigate regional disparities to a certain extent, this study employs per capita public budget expenditures on higher education as a metric to represent higher education levels across regions. This approach allows for a more effective demonstration of the variations in investment in higher education by local governments. The specific measurement methodology adopted in this study is:

$$Edu = \frac{Budget}{Number} \tag{3}$$

In this analysis, 'Edu' denotes the level of local higher education, as reflected by the per capita public budget expenditure on higher education. 'Budget' signifies the public fiscal allocation

**Table 2. Variable definitions and calculation methods.**

| Type | Name | Symbol | Calculation Method |
|------|------|--------|--------------------|
| DV | Carbon intensity | Carbon | Carbon Emissions / GDP |
| IV | Higher education | lnedu | Higher education per capita general public budget education expenses take logarithmic figures |
| CV | Economic development | lnpergdp | Natural Logarithm of Per Capita GDP |
| CV | Population density | Density | Population / Area |
| CV | Urbanization | Urban | Urban Population / Total Population |
| CV | Transportation | lncar | Logarithm of Privately Owned Cars |
| CV | Trade openness | Open | (Imports + Exports) / Regional GDP |
| CV | Social consumption | Consump | Total Retail Sales of Consumer Goods/GDP |
| CV | Energy Structure | Energy | Regional Electricity Consumption/National Electricity Consumption |
| CV | Infrastructure | lnmile | Logarithm of Road Length |

for higher education, while 'Number' indicates the the number of students enrolled in local general colleges and universities. Data pertaining to the public fiscal allocation for higher education and university enrollments are obtained from the China Education Expenditure Statistical Yearbook.

**Control variables.** This study incorporates several control variables (CV), including economic development [58], population density [59], population, urbanization [60], transportation [61], trade openness [62], social consumption [63], energy structure [64], and infrastructure [65]. Economic development is represented by the natural logarithm of per capita GDP [66]. Population density is indicated by the ratio of regional year-end population to area. Urbanization is represented by the ratio of urban population to total population [67,68]. Transportation are represented by the natural logarithm of the number of privately owned cars. Trade openness is presented as the ratio of total imports and exports to regional GDP [69]. Social consumption is given by the ratio of total retail sales of consumer goods to GDP [70]. Energy structure is measured as the ratio of regional electricity consumption to national electricity consumption [71]. Infrastructure is measured by the natural logarithm of regional road mileage [72]. The description of each variable is presented in Table 2.

## Fixed effects model

To study the causal relationship between higher education and carbon intensity, it is necessary to establish an appropriate and effective econometric model. Employing F-tests, BP-tests, and Hausman tests, this study adopts a two-way fixed effects model as follows:

$$Carbon_{it} = \beta_0 + \beta_1 lnedu_{it} + \beta_3 control_{it} + \mu_i + \rho_t + \varepsilon_{it} \tag{4}$$

Where region (i) represents the geographical area, year (t) denotes the specific time period, $Carbon_{it}$ signifies the carbon intensity of region (i) at time (t), the term 'lnedu$_{it}$' represents the state of higher education in region i at time t. In this paper, it is indicated by the per capita general public budget expenditure on higher education, and control$_{it}$ encompasses a set of control variables that may influence carbon intensity. These variables include economic development (lnpgdp) [73], population density (density) [59], urbanization (urban) [60], transportation (lncar) [74], trade openness (open) [62], social consumption (consump) [63], energy structure (energy) [64], and infrastructure (lnmile) [65], as depicted in Table 3. To address heteroskedasticity, variables with significant value gaps undergo logarithmic transformation. The fixed effect, denoted by ui, accounts for underlying disparities in carbon intensity across different geographic locations, while the time fixed effect $\rho_t$ allows for flexible control over potential

**Table 3. Descriptive statistics of variables.**

| Variable | Obs | Mean | SD | Min | Max |
|---|---|---|---|---|---|
| Carbon | 600 | 3.83 | 4.88 | 0.38 | 40.98 |
| lnedu | 600 | 9.17 | 0.73 | 7.57 | 11.10 |
| lnpergdp | 600 | 10.28 | 0.85 | 7.97 | 12.12 |
| Density | 600 | 8.17 | 0.75 | 6.26 | 9.45 |
| Urban | 600 | 0.53 | 0.15 | 0.24 | 0.90 |
| lncar | 600 | 4.90 | 1.41 | 0.76 | 7.73 |
| Open | 600 | 0.31 | 0.38 | 0.01 | 1.84 |
| Consump | 600 | 0.36 | 0.63 | 0.22 | 0.54 |
| Energy | 600 | 0.03 | 0.02 | 0.01 | 0.11 |
| lnmile | 600 | 11.40 | 0.90 | 8.71 | 12.89 |

disparities across different time points. The error term $\varepsilon_{it}$ includes: 1) other insignificant influencing factors; 2) observational errors; 3) stochastic factors that are difficult to control and measure. Empirical data for this study spans from 2001 to 2020, covering a balanced panel dataset of 30 provincial-level administrative regions in China. The primary data sources include the "China Statistical Yearbook," as well as provincial and municipal statistical yearbooks and statistical bulletins. Data for Tibet, Hong Kong, Macau, and Taiwan are excluded due to data unavailability. Missing data points are filled using linear interpolation and ARIMA modeling. Table 3 presents the descriptive statistics of each variable.

## Temporal and spatial characteristics of carbon intensity

Fig 3 illustrates the distribution of carbon intensity across various regions of China in 2010 and 2020, with white indicating missing data for Hong Kong, Macau, Taiwan, and Tibet. It is evident that the eastern regions exhibit lower carbon intensity, while the central and western regions demonstrate higher carbon intensity. This pattern may be attributed to economic development and energy structures. On one hand, the eastern regions possess higher economic development and relatively higher per capita income, facilitating economic structural transformation and promoting green, low-carbon development. Conversely, the central and western regions experience delayed economic development and weaker economic foundations, resulting in less investment in resource utilization and environmental protection, thus leading to higher carbon intensity. On the other hand, the eastern regions have better access to natural gas and clean energy sources, while the central and western regions, due to geographical limitations and economic development, predominantly rely on traditional fossil fuels such as coal, leading to higher energy consumption and carbon emissions in these regions.

Fig 4 presents a bubble chart for China's thirty provincial-level administrative regions in 2010 (Left) and 2020 (Right), with bubble size indicating per capita GDP. This chart provides a more intuitive visualization of the relationship between higher education and carbon intensity, as well as the impact of economic development. It is evident that economic development plays a pivotal role in higher education and carbon intensity; regions with higher economic development also exhibit higher levels of higher education and lower carbon intensity.

Based on data from 2001 to 2020 for all 30 provinces in China, the upper and lower quartiles, maximum and minimum values, as well as the median of carbon intensity are calculated and presented in a boxplot (Fig 5). From this plot, it can be observed that, on one hand, significant variation exists in carbon intensity among different provinces. The highest degree of dispersion in carbon intensity is observed in 2001, indicating pronounced regional disparities in pollution levels. On the other hand, with the passage of time, the overall dispersion of the

**Fig 3.** Distribution of Carbon Intensity across Chinese Regions in 2010 (Left) and 2020 (Right).

comprehensive environmental pollution index among provinces tends to decrease, and the upper limit exhibits a fluctuating downward trend, suggesting a general decline in carbon intensity across regions.

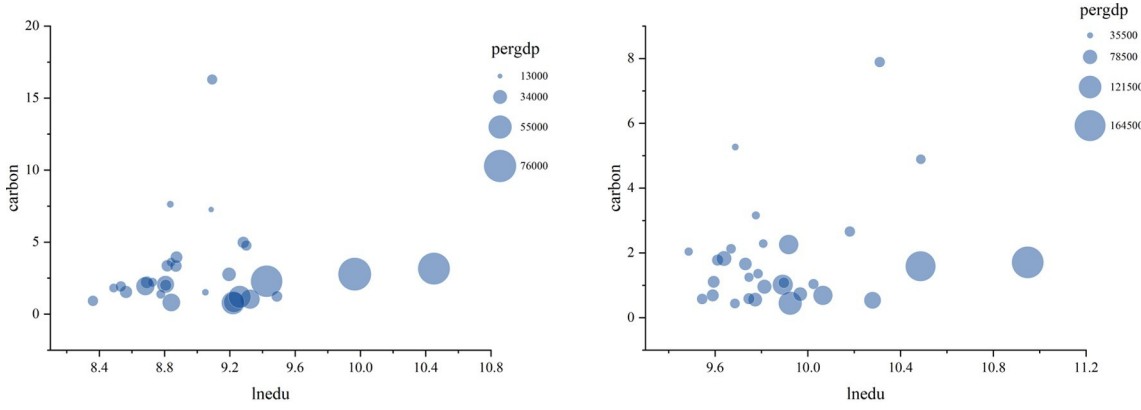

**Fig 4.** Bubble Chart of China's Provincial Regions in 2010 (Left) and 2020 (Right).

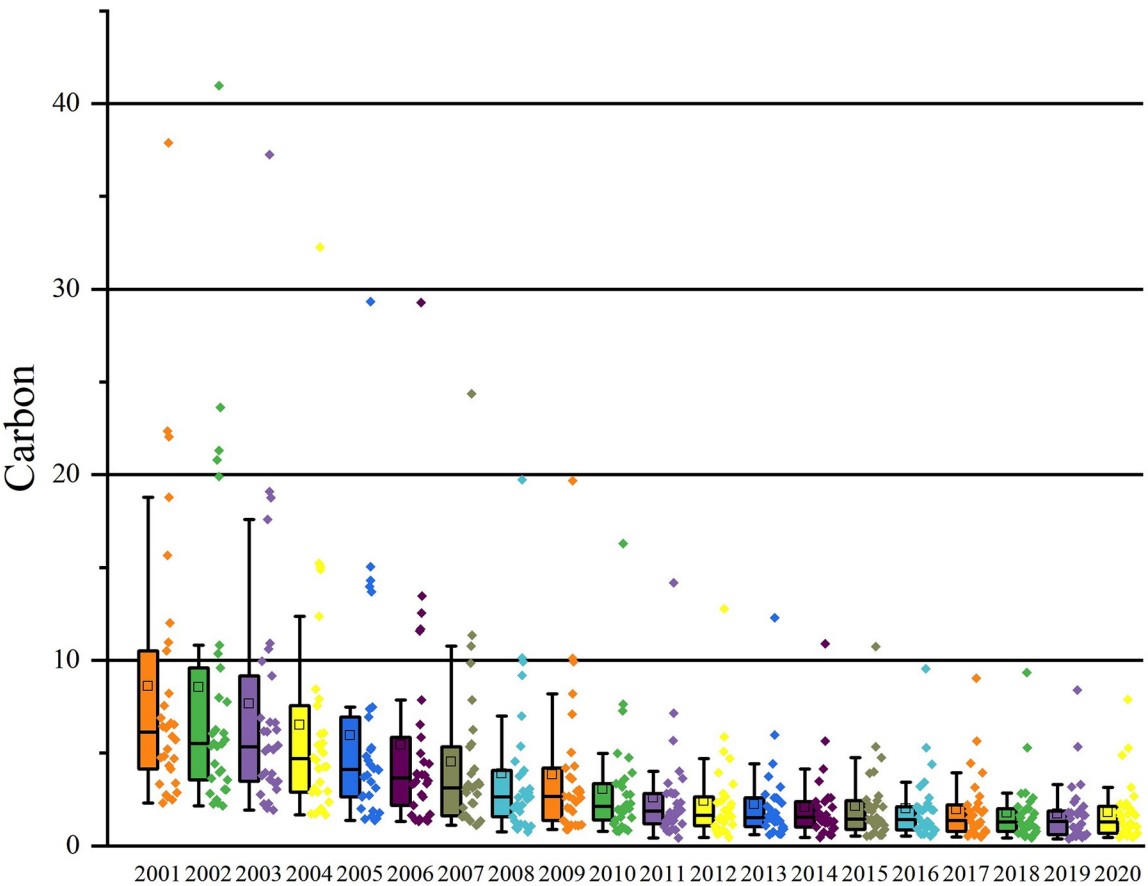

**Fig 5. Boxplot of carbon Intensity across Chinese regions from 2001 to 2020.**

Fig 6 illustrates the temporal evolution characteristics of higher education and carbon intensity in the 30 provincial-level administrative regions of China, as well as their dynamic relationship. Over the period of twenty years, from 2001 to 2020, there has been a continuous increase in higher education investment, accompanied by a gradual decrease in carbon intensity. It is worth noting that during the period from 2001 to 2010, which coincided with the first decade of expanded enrollment in Chinese higher education institutions, several regions, including Inner Mongolia, Beijing, Jilin, Sichuan, Ningxia, Anhui, and others, experienced a significant rise in higher education investment. This expansion provided greater opportunities for students to pursue higher education. Simultaneously, carbon intensity across regions continued to decline, indicating an inverse relationship between higher education investment and carbon intensity. However, during the subsequent decade from 2011 to 2020, the growth of higher education investment was more gradual, and carbon intensity reduction also followed a similarly moderate trend.

## 4. Empirical analysis

### Estimated results of fixed effects model

In this study, the Hausman test on panel data was conducted using STATA 17. The results indicate a p-value below 0.01, rejecting the null hypothesis of random effects. Subsequently,

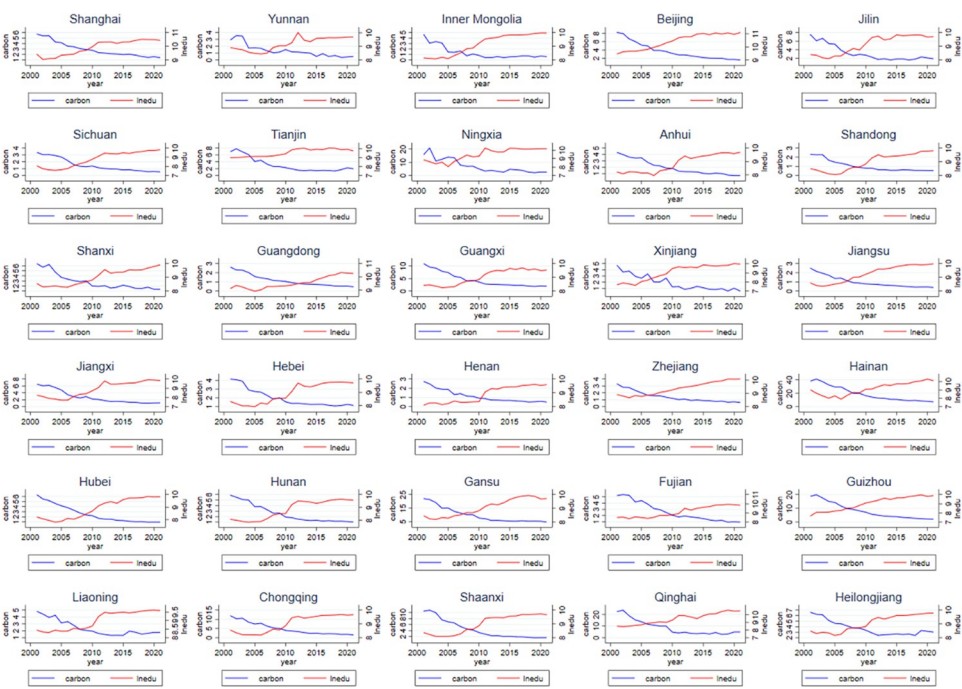

**Fig 6. Temporal trends of higher education and carbon intensity.**

annual dummy variables were introduced to assess the presence of individual time effects. The results also reveal a p-value below 0.01, strongly rejecting the null hypothesis of "no time effects," confirming the existence of time effects in the model. As a result, a two-way fixed effects model is chosen to explore the impact of higher education on carbon intensity. For comparative purposes, this model incorporates the mixed effects model and random effects model. Table 4 presents the regression results for (1) mixed effects model (OLS), (2) random effects model (RE), and (3) two-way fixed effects model (FE).

In general, upon observing the regression results in Table 4, the coefficient of determination R2 for the mixed-effects model in Model 1 is 0.659, for the random-effects model in Model (2) is 0.621, and for the two-way fixed-effects model in Model (3) is 0.921. This indicates that the two-way fixed effects model exhibits the best overall fit. Notably, from the regression results in the column (3) of Table 5, the core explanatory variable, higher education, exhibits a negative coefficient. Higher education demonstrates a significant negative correlation with carbon intensity. On average, for every 1% increase in the logarithmically transformed per capita higher education investment, carbon intensity in a given provincial-level administrative region decreases by 0.219%. This result is statistically significant at the 1% level and hypothesis 1 is tested. The empirical findings suggest that higher education investment significantly suppresses regional carbon intensity. China can enhance production efficiency in various sectors and resource utilization by increasing higher education investment, thus reducing carbon emissions and promoting ecological environment improvement.

With respect to the CV, the regression coefficients for the economic development (lnpgdp) and population density (density) are negative and both statistically significant at the 1% level. On average, carbon intensity decreases with higher economic development and population density. Regression coefficients for urbanization and energy consumption are positive, with urbanization being statistically significant at the 1% level and energy consumption at the 5% level. Carbon intensity increases with higher levels of urbanization and energy consumption.

**Table 4. Regression results.**

| Varible | OLS | RE | FE |
|---|---|---|---|
| | (1) | (2) | (3) |
| lnedu | -0.427** | -0.183* | -0.219*** |
| | (0.190) | (0.103) | (0.085) |
| lnpgdp | -0.207 | -0.562*** | -1.213*** |
| | (0.274) | (0.200) | (0.164) |
| Density | -0.460** | -0.526*** | -0.872*** |
| | (0.17) | (0.147) | (0.335) |
| Urban | -0.239 | 0.253 | 0.634*** |
| | (0.157) | (0.276) | (0.181) |
| lncar | -0.318 | -0.093 | -1.161 |
| | (0.252) | (0.329) | (0.207) |
| Open | -0.087 | -0.183** | -0.081 |
| | (0.071) | (0.074) | (0.087) |
| Consump | 0.152*** | 0.069* | 0.049 |
| | (0.053) | (0.040) | (0.036) |
| Energy | 0.253** | 0.184** | 0.260** |
| | (0.110) | (0.083) | (0.120) |
| lnmile | -0.144 | -0.048 | 0.222 |
| | (0.137) | (0.261) | (0.146) |
| Constant | 0.006 | -0.009 | -2.107*** |
| | (0.08) | (0.095) | (0.355) |
| Time FE | NO | NO | YES |
| Province FE | NO | NO | YES |
| N | 600 | 600 | 600 |
| $R^2$ | 0.659 | 0.621 | 0.921 |

Note: Standard errors are shown in parentheses; *, ** and *** indicate the significance at the 10%, 5% and 1% levels, respectively.

**Table 5. Regression results with alternative variables.**

| Variable | carbon | carbon |
|---|---|---|
| | (1) | (2) |
| eduyear | -0.068*** | |
| | (0.094) | |
| eduratio | | -0.872*** |
| | | (0.163) |
| Control | YES | YES |
| Time FE | YES | YES |
| Province FE | YES | YES |
| N | 600 | 600 |
| $R^2$ | 0.769 | 0.866 |

Note: Standard errors are shown in parentheses; *, ** and *** indicate the significance at the 10%, 5% and 1% levels, respectively.

Other factors such as transportation, trade openness, social consumption, and infrastructure development also exert certain impacts on carbon intensity, but these impacts are not statistically significant.

## Robustness tests

**Variable replacement method.** To verify the reliability of the conclusions and avoid incidental findings due to the selection of specific variables, this paper plans to change the core explanatory variable of higher education. Since higher education is defined from the perspective of input by the per capita general public budget expenditure on education, This paper selects the average years of education per capita and the proportion of the population with higher education as proxy variables for higher education from the perspective of higher education output. The calculation method for the average years of education per capita is shown in the formula, and the proportion of the population in higher education is represented by Formula; the proportion of the population in higher education is also expressed by Formula 6:

$$Eduyear = \frac{Primary*6 + junior*9 + high*12 + 16*higher}{Number} \tag{5}$$

$$Eduratio = \frac{higher}{Number} \tag{6}$$

Eduyear represents the average years of education per capita, primary denotes the number of individuals with primary school education, junior indicates the number of individuals with junior high school education, high stands for the number of individuals with high school education, higher signifies the number of individuals with education beyond college level, and number refers to the population aged six and above. The data are sourced from the China Statistical Yearbook and China Population Census data. The regression results, after substituting the higher education variable, are presented in Table 5. It can be observed that even after utilizing average years of education per capita and the proportion of the population with higher education as alternative indicators for higher education, the coefficients remain significantly negative. While there are slight disparities in parameter estimates between the alternative and core explanatory variables, the signs of the estimates remain unchanged, and they remain statistically significant at the 1% level. The goodness of fit for the models is also similar, indicating that both an increase in average years of education per capita and a higher proportion of the population with higher education lead to a reduction in local carbon intensity, which validates Hypothesis 1 from a tertiary education output perspective. Therefore, the conclusion that higher education can mitigate carbon intensity retains robustness.

**Heterogeneity analysis.** The economic landscape in our nation exhibits a stratified progression from the eastern coastal areas to the western inland regions. Variations in natural and cultural settings may influence the efficacy of higher education. This study categorizes the entire sample into eastern and central-western regions based on the classification method employed by the National Bureau of Statistics. Regression analysis, detailed in Table 6, demonstrates that higher education significantly reduces carbon intensity across both regions, with a more pronounced effect in the central-western region—a finding corroborated by the intergroup coefficient difference test. This disparity may stem from the poorer geographical and economic conditions of the central and western regions, making their energy structure and technological innovation more reliant on the advancement of higher education. This also proves that higher education indeed helps the relatively underdeveloped central and western regions reduce carbon intensity.

**Table 6. Heterogeneity analysis.**

| Varible | Eastern region | Central and Western region |
|---|---|---|
| | (1) | (2) |
| lnedu | -0.318* | -0.827*** |
| | (0.094) | (0.129) |
| Control | YES | YES |
| Time FE | YES | YES |
| Province FE | YES | YES |
| N | 220 | 380 |
| $R^2$ | 0.724 | 0.891 |
| Test for difference in coefficients between groups for lnedu | 0.047*** | |

Note: Standard errors are in parentheses; ***, **, and * represent significance levels of 1%, 5%, and 10%, respectively; empirical p-value results for the test of coefficient of variation between groups were obtained by bootstrap sampling 1000 times.

## 5. Mechanism analysis

Previous research has conclusively shown that enhancing higher education significantly reduces carbon intensity. However, the mechanisms through which higher education influences carbon intensity warrant further exploration. This study proposes three primary mechanisms: the dissemination of environmental consciousness, the upgrading of industrial structures, and the refinement of these structures, as discussed in the prior theoretical analysis. To empirically validate these mechanisms, this study employs a mediation analysis approach using a fixed effects model, drawing on the methodologies outlined by Baron [75] and Hayes [76]. We utilize the fixed effects model to develop the following three recursive equations:

$$Carbon_{it} = \alpha_{it} + \beta_1 lnedu_{it} + \delta control_{it} + u_i + \rho_t + \varepsilon_{it} \tag{7}$$

$$W_{it} = \alpha_{it} + \gamma_1 lnedu_{it} + \varphi control_{it} + u_i + \rho_t + \varepsilon_{it} \tag{8}$$

$$Carbon_{it} = \alpha_{it} + \beta_2 lnedu_{it} + \theta W_{it} + \mu control_{it} + u_i + \rho_t + \varepsilon_{it} \tag{9}$$

Herein, Carbon$_{it}$ represents the carbon intensity of region i at time t. lnedu$_{it}$ stands for the per capita public fiscal expenditure on higher education for region i at time t after taking the absolute value. control$_{it}$ is a set of control variables that may affect carbon intensity, including economic development [73], population density [59], urbanization [60], transportation [74], trade openness [62], social consumption [63], energy structure [64], and infrastructure [65]. W$_{it}$ serves as the mediator variable, including: (1) technological innovation investment, represented by the logarithm of research and development expenditure (rd) [44]; (2) public environmental awareness, represented by the completed investment in industrial pollution control [77]; and (3) industrial structure, represented by the ratio of tertiary industry output value to secondary industry output value. The steps for examining the mediation effects are as follows [75,76]: (1) Estimate Model (6) to determine if higher education affects carbon intensity. If β1 is significantly negative, it indicates that higher education has an inhibitory effect on carbon intensity. This has already been demonstrated in the table. (2) Estimate Model (7) to investigate the relationship between higher education and the mediator variable. If it is significantly positive, it suggests that higher education promotes the mediator variable; if negative, it implies an inhibitory effect. (3) Estimate Model (8). If at least one of the coefficients θ is insignificant, further testing is needed. If both θ coefficients are significant, a mediation effect

**Table 7. Results of the intermediary effect test.**

| Varible | Technological Innovation | | Environmental Awareness | | Industrial structure | |
|---|---|---|---|---|---|---|
| | (1) | (2) | (3) | (4) | (5) | (6) |
| | RD | Carbon | Pollution | Carbon | Industrial | Carbon |
| lnedu | 0.382** | -0.150* | 0.254* | -0.162** | 0.294** | -0.146* |
| | (0.184) | (0.077) | (0.211) | (0.083) | (0.055) | (0.085) |
| RD | | -0.477*** | | | | |
| | | (0.042) | | | | |
| Pollution | | | | -0.186*** | | |
| | | | | (0.034) | | |
| Industrial | | | | | | -0.202*** |
| | | | | | | (0.066) |
| Constant | 0.173 | -0.719*** | -0.572 | -1.532*** | -1.326*** | -1.953*** |
| | (0.293) | (0.290) | (0.392) | (0.315) | (-0.206) | (0.355) |
| Control | YES | YES | YES | YES | YES | YES |
| Time FE | YES | YES | YES | YES | YES | YES |
| Province FE | YES | YES | YES | YES | YES | YES |
| N | 600 | 600 | 600 | 600 | 600 | 600 |
| $R^2$ | 0.955 | 0.937 | 0.760 | 0.815 | 0.977 | 0.811 |

Note: Standard errors are shown in parentheses; *, ** and *** indicate the significance at the 10%, 5% and 1% levels, respectively.

exists. If the regression coefficient β2 is also significant, it indicates partial mediation effect; if β2 is insignificant, it suggests complete mediation effect. Table 7 reports the results of the second and third stages of the mediation effect test. Columns (1), (3), and (5) confirm that higher education significantly influences technology innovation, environmental awareness, and industrial structure. This signals readiness for the subsequent mediation effect examination. Columns (2), (4), and (6) present the results of the third-stage estimation of the mediation effect model, testing the impact of technology innovation, environmental awareness, and industrial structure on carbon intensity. In each case, the coefficients are significantly negative, indicating that technology innovation, environmental awareness, and industrial structure all exhibit partial mediation effects. Therefore, higher education can achieve the goal of reducing carbon intensity through mechanisms such as technological innovation incentives, the popularization of environmental awareness, and the upgrading of industrial structures, with Hypotheses 2, 3, and 4 being validated.

# 6. Conclusion and recommendations

## Conclusion

Using a panel dataset comprising 30 provinces in China spanning the years 2001 to 2020, this study utilizes a fixed effects model to examine the influence of higher education on carbon intensity. In doing so, the paper elucidates the underlying mechanisms by which higher education contributes to the mitigation of carbon intensity. The primary conclusions are proffered as follows: Firstly, an evident negative correlation is observed between higher education and carbon intensity. On average, a logarithmic increase of 1% in per capita higher education expenditure corresponds to a reduction of 0.219% in carbon intensity within each provincial administrative region of China. This result attains statistical significance at the 1% level. Secondly, When replacing higher education with variables such as the average years of education per capita and the proportion of the population with higher education, the analysis reveals that

increases in both the average years of education per capita and the proportion of the population with higher education lead to a reduction in local carbon intensity. This further substantiates the conclusion that higher education leads to a reduction in regional carbon intensity. Thirdly, this paper divides China into eastern and central-western regions for a heterogeneity analysis, which reveals that higher education significantly affects carbon intensity in both regions, with a more pronounced impact in the central-western region. Fourthly, by employing a mediation effect model, this paper exhibits that higher education can accomplish the objective of reducing carbon intensity through mechanisms such as technology innovation incentive, environmental awareness dissemination, and industrial structural upgrading.

## Policy recommendations

Firstly, the amelioration of higher education can assuage carbon intensity. Universities can integrate carbon consumption reduction and sustainable lifestyles into their curriculum, proffering pertinent specialized or elective courses to inculcate the concept and skills of low-carbon living in students. By forging partnerships with enterprises, government agencies, and pertinent social organizations, universities can orchestrate students' involvement in environmental activities and public lectures to propagate the notion of low-carbon living in society, thereby augmenting public awareness and responsibility towards environmental issues and, thereby, curtailing regional carbon emissions. Secondly, universities bear the onus of researching and developing low-carbon technologies, involving renewable energy, energy-saving technology, and carbon capture and storage. The government can offer financial support and incentive measures to universities, thereby encouraging them to undertake research projects germane to the environment and carbon reduction. Innovative accomplishments can be rewarded with financial backing, fostering breakthroughs in carbon reduction research. Finally, considering the regional heterogeneity of higher education's impact on China's carbon intensity, particularly its pronounced effect in the central and western regions, it is incumbent upon the government to devise targeted environmental policies. The government can facilitate information exchange, resource sharing, and collaboration between universities and enterprises through collaborative platforms, while concurrently establishing platforms for technology transfer and industrialization support. These endeavors can support higher education institutions in translating research advancements into tangible production, fostering the dissemination and implementation of green technology, and facilitating the transition and enhancement of conventional industries towards low-carbon sectors.

## Supporting information

**S1 Data.**
(XLSX)

## Acknowledgments

Thanks to the judging experts and all members of our team for their insightful advice.

## Author Contributions

**Conceptualization:** Qin Yuan, Ruiqi Wang, Huanchen Tang.

**Data curation:** Ruiqi Wang, Huanchen Tang.

**Formal analysis:** Ruiqi Wang, Huanchen Tang.

**Investigation:** Ruiqi Wang, Huanchen Tang.

**Methodology:** Ruiqi Wang, Huanchen Tang.

**Project administration:** Ruiqi Wang.

**Resources:** Qin Yuan, Huanchen Tang.

**Software:** Ruiqi Wang.

**Supervision:** Qin Yuan, Xin Ma.

**Validation:** Huanchen Tang.

**Visualization:** Qin Yuan.

**Writing – original draft:** Ruiqi Wang, Huanchen Tang.

**Writing – review & editing:** Qin Yuan, Huanchen Tang, Xin Ma, Xinyue Zeng.

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
