## [Decision Letter · Decision Letter 0]

19 Jun 2024

PONE-D-24-17034A Study on the Potential of Higher Education in Reducing Carbon IntensityPLOS ONE

Dear Dr. tang,

Thank you for submitting your manuscript to PLOS ONE. After careful consideration, we feel that it has merit but does not fully meet PLOS ONE’s publication criteria as it currently stands. Therefore, we invite you to submit a revised version of the manuscript that addresses the points raised during the review process.

Please supplement the theoretical framework and optimize the literature review.The rationality of methods (assumptions, calculation formulas, frameworks) needs to be further clarified.Clarify the conclusion of this paper, which should be specific and targeted. ==============================

We look forward to receiving your revised manuscript.

Kind regards,

Bifeng Zhu

Academic Editor

PLOS ONE

2. In the online submission form you indicate that your data is not available for proprietary reasons and have provided a contact point for accessing this data. Please note that your current contact point is a co-author on this manuscript. According to our Data Policy, the contact point must not be an author on the manuscript and must be an institutional contact, ideally not an individual. Please revise your data statement to a non-author institutional point of contact, such as a data access or ethics committee, and send this to us via return email. Please also include contact information for the third party organization, and please include the full citation of where the data can be found.

3. We note that Figure 2 in your submission contain [map/satellite] images which may be copyrighted. All PLOS content is published under the Creative Commons Attribution License (CC BY 4.0), which means that the manuscript, images, and Supporting Information files will be freely available online, and any third party is permitted to access, download, copy, distribute, and use these materials in any way, even commercially, with proper attribution. For these reasons, we cannot publish previously copyrighted maps or satellite images created using proprietary data, such as Google software (Google Maps, Street View, and Earth). For more information, see our copyright guidelines: http://journals.plos.org/plosone/s/licenses-and-copyright.

1. You may seek permission from the original copyright holder of Figure 2 to publish the content specifically under the CC BY 4.0 license.  

Additional Editor Comments (if provided):

Reviewers' comments:

Reviewer's Responses to Questions

**Comments to the Author**

1. Is the manuscript technically sound, and do the data support the conclusions?

Reviewer #1: Yes

Reviewer #2: Yes

2. Has the statistical analysis been performed appropriately and rigorously? 

Reviewer #1: Yes

Reviewer #2: Yes

3. Have the authors made all data underlying the findings in their manuscript fully available?

Reviewer #1: Yes

Reviewer #2: Yes

4. Is the manuscript presented in an intelligible fashion and written in standard English?

Reviewer #1: Yes

Reviewer #2: Yes

5. Review Comments to the Author

Reviewer #1: Using a panel dataset comprising 30 provinces in China, the author utilizes a fixed effects model to examine the influence of higher education on carbon intensity. Although the attractiveness of the subject matter, the paper suffers from serious problems. For the following reasons, we suggest this study to be appropriated with major revision.

1. The marginal contribution of this paper does not seem to be clear, please add the marginal contribution in the introduction section.

2. The theoretical aspect of this article is relatively insufficient. I believe that literature review cannot replace theoretical analysis. I suggest that the author add a section on theoretical analysis and improve the research hypothesis.

3. In the Empirical Analysis section, the author failed to control for time fixed effects and province fixed effects in benchmark regression and heterogeneity analysis, which is crucial for the accuracy of the estimation. Suggest the author to re estimate and control for time fixed effects and province fixed effects.

4. Heterogeneity analysis lacks coefficient difference testing and cannot directly compare sizes.

Reviewer #2: This article explores the relationship between carbon emissions and higher education, offering a unique and innovative perspective. However, several areas require improvement and further clarification:

1. The author examined the impact of higher education on carbon emissions. However, environmental awareness education at the primary or secondary school levels also affects carbon emissions. How did the author account for the potential interference from primary and secondary education on carbon emissions?

2. In the Introduction section, the author explains the relationship between carbon emissions and higher education, but the discussion lacks depth. It is recommended that the author further elaborate on the relationship between higher education and carbon emissions by integrating existing research. Additionally, the author should consider reducing some of the common knowledge content in the Introduction.

3. The author has obtained some quantitative conclusions, but these are not reflected in the abstract. It is recommended that the author include some quantitative conclusions in the abstract.

6. PLOS authors have the option to publish the peer review history of their article (what does this mean?). If published, this will include your full peer review and any attached files.

Reviewer #1: No

Reviewer #2: No

---

## [Author Response · Author response to Decision Letter 0]

20 Jul 2024

Dear Editor and Reviewers: 

We sincerely thank you and the reviewers for the very helpful comments and constructive suggestions that have led to significant improvements of our paper. We have carefully addressed your comments point-by-point in the revised manuscript. Specific responses are placed in an additional file for editors and reviewers.

The main changes and revisions include 1) We reworked the introduction section to further add the relationship between carbon emissions and higher education and to increase the marginal contribution of the article.; 2) We added 2.3 Theoretical analysis and research hypothesis to refine the research hypotheses; 3) In the empirical research section, we re-estimated and controlled for time fixed effects and province fixed effects, and reworked the heterogeneity analysis; 4) We reworked the abstract section to add quantitative results.

We hope that the changes and responses to comments in the revised manuscript are sufficient to make our manuscript suitable for publication in PLOS ONE.

Response to reviewer 1's comments

Using a panel dataset comprising 30 provinces in China, the author utilizes a fixed effects model to examine the influence of higher education on carbon intensity. Although the attractiveness of the subject matter, the paper suffers from serious problems. For the following reasons, we suggest this study to be appropriated with major revision.

Thank you very much for the reviewer's recognition of the topic and the following suggestions, we have carefully revised the article according to your suggestions, and the main changes are as follows: 1) We reworked the introduction section to further add the relationship between carbon emissions and higher education and to increase the marginal contribution of the article; 2) We added Theoretical analysis and research hypothesis to refine the research hypotheses; 3) In the empirical research section, we re-estimated and controlled for time fixed effects and province fixed effects, and reworked the heterogeneity analysis; 4) We reworked the abstract section to add quantitative results.

1. The marginal contribution of this paper does not seem to be clear, please add the marginal contribution in the introduction section.

Response 1: Many thanks to the reviewers for their suggestions, and we have added the marginal contribution of this paper in the Introduction section（Lines 77-88）.

This study analyzes the relationship between higher education and carbon intensity across thirty provincial administrative regions in China using a fixed-effects model. To strengthen the robustness of our findings, we substituted the core explanatory variables, further validating our conclusions. The analysis then categorizes China into eastern and central-western regions to explore regional disparities in the impact of higher education on carbon intensity. Additionally, we developed a mediation effect model that elucidates the roles of technological innovation incentives, environmental awareness promotion, and industrial structure enhancement in mediating this relationship. Based on these insights, we offer targeted policy recommendations. In comparison to existing studies, this paper makes three primary contributions: First, it proposes a theoretical framework from the perspective of higher education, thereby broadening the environmental governance discourse. Second, it addresses both the spatial variability in higher education and its differential impacts on carbon intensity, considering both educational inputs and outputs. Third, it identifies critical mechanisms such as technological innovation, environmental awareness, and industrial upgrading that link higher education and carbon intensity. 

2. The theoretical aspect of this article is relatively insufficient. I believe that literature review cannot replace theoretical analysis. I suggest that the author add a section on theoretical analysis and improve the research hypothesis.

Response 2: The reviewers' suggestions are greatly appreciated. We have re-added Theoretical analysis and research hypothesis to more systematically organize the theory and improve the research hypothesis of this paper（Lines 172-207）.

Theoretical analysis and research hypothesis

From a behavioral economics standpoint, highly educated individuals typically exhibit a heightened sense of social responsibility. They are more attentive to environmental sustainability and actively oppose lifestyles and consumption habits characterized by high energy use. Such individuals are predisposed to adopting pro-environmental behaviors and demonstrating a robust environmental consciousness. In both professional and personal contexts, they prefer to efficiently utilize idle resources, opt for low-carbon transportation solutions, and use environmentally friendly products. For example, as consumers, educated individuals tend to favor purchasing electric vehicles to mitigate carbon emissions43. Furthermore, educated entrepreneurs and managers are more likely to uphold corporate social responsibilities, advocate for sustainable production methods, and minimize the ecological impact of their business operations.

According to human capital theory, education and training represent crucial investments in human capital. Higher education not only augments workers' professional knowledge and practical skills but also fosters their capacity for innovation and research. This comprehensive enhancement significantly improves their level of knowledge, professional abilities, and overall competencies, thereby increasing their potential future earnings and job prospects. The educational advancement of the workforce contributes to the accumulation and diversification of human capital, which, in turn, drives the growth of high-tech industries and the evolution of industrial structures. Consequently, this shift leads to the displacement of traditional high-pollution industries by more sustainable high-tech and green sectors, effectively reducing carbon intensity.

Theoretical economics suggests that higher education not only raises an individual’s knowledge level but also generates knowledge spillover effects. This educational process not only benefits the learners themselves but also exerts a positive influence on the wider society. By engaging in both educational and research activities, universities can attract technology-driven enterprises to establish collaborative platforms such as joint laboratories, technology transfer centers, and innovation incubators. These platforms not only enable joint research projects and technological advancements but also facilitate the specialized training of innovative talent. They promote local technological innovation, catalyze the development of emerging clean technologies, improve resource efficiency in production, and ultimately contribute to a reduction in carbon intensity.

Based on the analysis above, this paper illustrates the discussed mechanisms in Figure 1 and proposes the following hypotheses: 

Hypothesis 1: Higher education can significantly suppress carbon intensity. 

Hypothesis 2: Higher education reduces carbon intensity through the mechanism of promoting environmental awareness. 

Hypothesis 3: Higher education reduces carbon intensity through the mechanism of upgrading industrial structures. 

Hypothesis 4: Higher education reduces carbon intensity through the mechanism of incentivizing technological innovation.

Fig 1. Mechanistic analysis of higher education and carbon intensity.

3. In the Empirical Analysis section, the author failed to control for time fixed effects and province fixed effects in benchmark regression and heterogeneity analysis, which is crucial for the accuracy of the estimation. Suggest the author to re estimate and control for time fixed effects and province fixed effects.

Response 3: For the empirical analysis section, we re-estimated and controlled for time-fixed effects and province-fixed effects in accordance with your recommendations（Lines 345-372）.

Estimated Results of Fixed Effects Model

In this study, the Hausman test on panel data was conducted using STATA 17. The results indicate a p-value below 0.01, rejecting the null hypothesis of random effects. Subsequently, annual dummy variables were introduced to assess the presence of individual time effects. The results also reveal a p-value below 0.01, strongly rejecting the null hypothesis of "no time effects," confirming the existence of time effects in the model. As a result, a two-way fixed effects model is chosen to explore the impact of higher education on carbon intensity. For comparative purposes, this model incorporates the mixed effects model and random effects model. Table 4 presents the regression results for (1) mixed effects model (OLS), (2) random effects model (RE), and (3) two-way fixed effects model (FE).

In general, upon observing the regression results in Table 4, the coefficient of determination R2 for the mixed-effects model in Model 1 is 0.659, for the random-effects model in Model (2) is 0.621, and for the two-way fixed-effects model in Model (3) is 0.921. This indicates that the two-way fixed effects model exhibits the best overall fit. Notably, from the regression results in the column (3) of Table 5, the core explanatory variable, higher education, exhibits a negative coefficient. Higher education demonstrates a significant negative correlation with carbon intensity. On average, for every 1% increase in the logarithmically transformed per capita higher education investment, carbon intensity in a given provincial-level administrative region decreases by 0.219%. This result is statistically significant at the 1% level and hypothesis 1 is tested. The empirical findings suggest that higher education investment significantly suppresses regional carbon intensity. China can enhance production efficiency in various sectors and resource utilization by increasing higher education investment, thus reducing carbon emissions and promoting ecological environment improvement.

With respect to the CV, the regression coefficients for the economic development (lnpgdp) and population density (density) are negative and both statistically significant at the 1% level. On average, carbon intensity decreases with higher economic development and population density. Regression coefficients for urbanization and energy consumption are positive, with urbanization being statistically significant at the 1% level and energy consumption at the 5% level. Carbon intensity increases with higher levels of urbanization and energy consumption. Other factors such as transportation, trade openness, social consumption, and infrastructure development also exert certain impacts on carbon intensity, but these impacts are not statistically significant.

Tab 4. Regression Results.

Varible OLS RE FE

 (1) (2) (3)

lnedu -0.427** -0.183* -0.219***

 (0.190) (0.103) (0.085)

lnpgdp -0.207 -0.562*** -1.213***

 (0.274) (0.200) (0.164)

Density -0.460** -0.526*** -0.872***

 (0.17) (0.147) (0.335)

Urban -0.239 0.253 0.634***

 (0.157) (0.276) (0.181)

lncar -0.318 -0.093 -1.161

 (0.252) (0.329) (0.207)

Open -0.087 -0.183** -0.081

 (0.071) (0.074) (0.087)

Consump 0.152*** 0.069* 0.049

 (0.053) (0.040) (0.036)

Energy 0.253** 0.184** 0.260**

 (0.110) (0.083) (0.120)

lnmile -0.144 -0.048 0.222

 (0.137) (0.261) (0.146)

Constant 0.006 -0.009 -2.107***

 (0.08) (0.095) (0.355)

Time FE NO NO YES

Province FE NO NO YES

N 600 600 600

R2 0.659 0.621 0.921

Note: Standard errors are shown in parentheses; *, ** and *** indicate the significance at the 10%, 5% and 1% levels, respectively.

4. Heterogeneity analysis lacks coefficient difference testing and cannot directly compare sizes.

Response 4: Based on your suggestions, we have reworked the heterogeneity analysis.

Heterogeneity Analysis

 The economic landscape in our nation exhibits a stratified progression from the eastern coastal areas to the western inland regions. Variations in natural and cultural settings may influence the efficacy of higher education. This study categorizes the entire sample into eastern and central-western regions based on the classification method employed by the National Bureau of Statistics. Regression analysis, detailed in Table 7, demonstrates that higher education significantly reduces carbon intensity across both regions, with a more pronounced effect in the central-western region—a finding corroborated by the inter-group coefficient difference test. This disparity may stem from the central-western region's less favorable geographical and economic conditions, which render its energy structure and technological innovation more responsive to advancements in higher education, thereby facilitating a reduction in carbon intensity. 

Tab 7. Heterogeneity Analysis.

Varible Eastern region Central and Western region

 (1) (2)

lnedu -0.318* -0.827***

 (0.094) (0.129)

Control YES YES

Time FE YES YES

Province FE YES YES

N 220 380

R2 0.724 0.891

Test for difference in coefficients between groups for lnedu 0.047***

Note: Standard errors are in parentheses; ***, **, and * represent significance levels of 1%, 5%, and 10%, respectively; empirical p-value results for the test of coefficient of variation between groups were obtained by bootstrap sampling 1000 times.

Response to reviewer 2's comments

This article explores the relationship between carbon emissions and higher education, offering a unique and innovative perspective. However, several areas require improvement and further clarification:

Thank you very much for the reviewer's recognition of the topic and the following suggestions, we have carefully revised the article according to your suggestions, and the main changes are as follows: 1) We reworked the introduction section to further add the relationship between carbon emissions and higher education and to increase the marginal contribution of the article.; 2) We added Theoretical analysis and research hypothesis to refine the research hypotheses; 3) In the empirical research section, we re-estimated and controlled for time fixed effects and province fixed effects, and reworked the heterogeneity analysis; 4) We reworked the abstract section to add quantitative results.

1. The author examined the impact of higher education on carbon emissions. However, environmental awareness education at the primary or secondary school levels also affects carbon emissions. How did the author account for the potential interference from primary and secondary education on carbon emissions?

Response 1: 为了排除小学或中学阶段的环保意识教育也会影响碳强度，本文分别从高等教育投入与高等教育产出视角选择自变量，其中高等教育投入角度本文选取高等教育方面的人均公共预算支出作为各地区高等教育水平的指标;高等教育产出角度本文选取人均教育年限和高等教育的人口比例作为高等教育的代理变量，考虑到目前中国已普及义务教育，因此人均教育年限和高等教育人口比例可以很好的反应中国高等教育产出情况。In order to rule out the influence of environmental awareness education at the elementary and secondary school levels on carbon intensity, this paper selects independent variables from the perspectives of higher education inputs and outputs. For the perspective of higher education inputs, this paper chooses the per capita public budget expenditure in higher education as an indicator of the level of higher education in various regions. From the perspective of higher education outputs, this paper selects the average years of education per capita and the proportion of the population with higher education as proxy variables for higher education. Considering that compulsory education has been universalized in China, the average years of education per capita and the proportion of the population with higher education can effectively reflect the outputs of higher education in China.

This approach allows for a more effective demonstration of the variations in investment in higher education by local governments. The specific measurement methodology adopted in this study is:

In this analysis, 'Edu' denotes the level of local higher education, as reflected by the per capita public budget expenditure on higher education. 'Budget' signifies the public fiscal allocation for higher education, while 'Number' indicates the the number of students enrolled in local general colleges and universities.. Data pertaining to the public fiscal allocation for higher education and university enrollments are obtained from th

---

## [Decision Letter · Decision Letter 1]

14 Aug 2024

A Study on the Potential of Higher Education in Reducing Carbon Intensity

PONE-D-24-17034R1

Dear Dr. tang,

We’re pleased to inform you that your manuscript has been judged scientifically suitable for publication and will be formally accepted for publication once it meets all outstanding technical requirements.

Kind regards,

Bifeng Zhu

Academic Editor

PLOS ONE

Additional Editor Comments (optional):

Reviewers' comments:

Reviewer's Responses to Questions

**Comments to the Author**

1. If the authors have adequately addressed your comments raised in a previous round of review and you feel that this manuscript is now acceptable for publication, you may indicate that here to bypass the “Comments to the Author” section, enter your conflict of interest statement in the “Confidential to Editor” section, and submit your "Accept" recommendation.

Reviewer #1: All comments have been addressed

2. Is the manuscript technically sound, and do the data support the conclusions?

Reviewer #1: Yes

3. Has the statistical analysis been performed appropriately and rigorously? 

Reviewer #1: Yes

4. Have the authors made all data underlying the findings in their manuscript fully available?

Reviewer #1: Yes

5. Is the manuscript presented in an intelligible fashion and written in standard English?

Reviewer #1: Yes

6. Review Comments to the Author

Reviewer #1: The author has fully addressed my concerns and hopes that this research can contribute to the relevant field.

7. PLOS authors have the option to publish the peer review history of their article (what does this mean?). If published, this will include your full peer review and any attached files.

Reviewer #1: No

---

## [Editor Report · Acceptance letter]

19 Aug 2024

PONE-D-24-17034R1 

PLOS ONE

Dear Dr. Tang, 

I'm pleased to inform you that your manuscript has been deemed suitable for publication in PLOS ONE. Congratulations! Your manuscript is now being handed over to our production team.

Kind regards, 

on behalf of

Dr. Bifeng Zhu 

Academic Editor

PLOS ONE